# Comparison of Various Smoothness Metrics for Upper Limb Movements in Middle-Aged Healthy Subjects

**DOI:** 10.3390/s23031158

**Published:** 2023-01-19

**Authors:** Nicolas Bayle, Mathieu Lempereur, Emilie Hutin, Damien Motavasseli, Olivier Remy-Neris, Jean-Michel Gracies, Gwenaël Cornec

**Affiliations:** 1UR 7377 BIOTN, Paris Est Créteil University (UPEC), F-94000 Créteil, France; 2AP-HP, Service de Rééducation Neurolocomotrice, Unité de Neurorééducation, Hôpitaux Universitaires Henri Mondor, F-94000 Créteil, France; 3U1101 LaTIM, Brest University, F-29200 Brest, France; 4Neurological Physical Medicine and Rehabilitation Department, University Hospital of Brest, F-29200 Brest, France

**Keywords:** motion analysis, kinematics, pointing task, smoothness metrics, SPARC, LDLJ

## Abstract

Backgound: Metrics for movement smoothness include the number of zero-crossings on the acceleration profile (N0C), the log dimensionless jerk (LDLJ), the normalized averaged rectified jerk (NARJ) and the spectral arc length (SPARC). Sensitivity to the handedness and movement type of these four metrics was compared and correlations with other kinematic parameters were explored in healthy subjects. Methods: Thirty-two healthy participants underwent 3D upper limb motion analysis during two sets of pointing movements on each side. They performed forward- and backward-pointing movements at a self-selected speed to a target located ahead at shoulder height and at 90% arm length, with and without a three-second pause between forward and backward movements. Kinematics were collected, and smoothness metrics were computed. Results: LDLJ, NARJ and N0C found backward movements to be smoother, while SPARC found the opposite. Inter- and intra-subject coefficients of variation were lowest for SPARC. LDLJ, NARJ and N0C were correlated with each other and with movement time, unlike SPARC. Conclusion: There are major differences between smoothness metrics measured in the temporal domain (N0C, LDLJ, NARJ), which depend on movement time, and those measured in the frequency domain, the SPARC, which gave results opposite to the other metrics when comparing backward and forward movements.

## 1. Introduction

Quantitative assessment of upper limb mobility [1,2,3] often uses clinical (using manual goniometry) or instrumented (using accelerometers, electronic goniometers, or 3D motion analysis tools) evaluation of segmental displacements (passive and active range of motion) [4]. These assessments, basically based on position, velocity and acceleration calculation, may lack sensitivity to detect small changes in movement trajectories and velocities in slowly progressive disorders such as spastic paresis syndromes or parkinsonism [2]. They may be inadequate to differentiate between various types of movement slowness (bradykinesia) induced by the many kinds of neurological, orthopedic and psychological disorders (as parkinsonian bradykinesia and voluntary or drug-induced bradykinesia) [5,6] or to demonstrate changes after rehabilitation programs or injections of blocking agents in chronic neurological disorders [7].

In healthy individuals, most discrete reach-to-point movements use a single acceleration burst per movement, making the resulting movement “smooth” with a bell-shaped velocity profile; these movements may be modeled as successions of sub-movements closely overlapping at each instant [8,9,10]. Most human motor disorders, by introducing irregularities in the trajectories and velocity profiles, make the resulting movements less smoothly, with more movement interruptions [10,11]. As defined by Balasubramanian, “movement smoothness is a quality related to the continuality or non-intermittency of a movement, independent of its amplitude and duration” [12]. Thus, movement smoothness metrics may represent an alternative to the classic clinical tools to better assess small changes in movement kinematic properties [12]. Several mathematical metrics to quantify smoothness have been suggested over time [12,13,14]. Some of those metrics are based on the observation of changes in the kinematic properties of movements such as changes in movement trajectories, changes in the velocity profile or assessments of the rates of changes in the acceleration profile (jerk-derived metrics) [9,10,11,14]. Those metrics of smoothness can be categorized as using the “temporal domain” and have been criticized as having shortcomings such as strong sensitivity to measurement noise and direct dependence to other kinematic parameters (movement amplitude and duration), lack of reliability and poor robustness [12,13]. Other smoothness metrics based on the detection of changes in frequencies of movement components have been more recently developed, considering that smooth movements contain a less complex frequency spectrum, in contrast with unsmooth movements, which may be contaminated by diverse frequency components [12,13]. Among the latter metrics, the spectral arc length measure (SPARC), quantifies the complexity of the Fourier magnitude spectrum of the velocity profile; it can be categorized using the “frequency domain” [12,13].

Comparisons of these various smoothness metrics across clinical protocols are difficult [6,15]. Validations and comparisons of these metrics have been mostly performed on simulated movements [12,13]. In a recent study, Refai et al. (2021) assessed 32 smoothness metrics used in stroke patients with mathematical criteria to be accepted as smoothness measures (metrics had to be dimensionless, reproducible, based on rate of change of position and not be a linear transform of other smoothness metrics), then tested them for their response to simulated changes in reaching [16]. They ended up recommending SPARC as a valid smoothness metric in both reach-to-point and reach-to-grasp tasks of the upper limb after stroke.

It thus appears that comparisons of the properties of the main smoothness metrics for upper limb (UL) movements of interest are required, not based on simulations but on actual human data (i.e., movements commonly used in neuro-rehabilitation programs to assess changes, such as forward- or sideways-pointing movements), acquired from middle-aged and seniors healthy participants (ages of onset of various neurodegenerative disorders, and most of the time age of the patients in our rehabilitation units), to better interpret future patient data, especially in patients with neurological disorders such as stroke or parkinsonism, and to demonstrate changes after rehabilitation programs.

This study aimed to measure healthy values and compare the properties of the SPARC, the normalized average rectified jerk (NARJ), the log dimensionless jerk (LDLJ) and the number of zero-crossings in the acceleration profile (N0C) for point-to-reach movements in healthy subjects.

## 2. Materials and Methods

This prospective study was conducted in accordance with the Declaration of Helsinki (2008), Good Clinical Practice guidelines and local regulatory requirements (registration number, ID-RCB: NCT01383512). It was approved by the Brest University Hospital IRB (n°653). All subjects gave written consent to the inclusion of material pertaining to them. All the experimental sessions were conducted in December 2019 in the Laboratory of Movement Analysis of Brest University Hospital (Brest, France). We recruited 32 healthy participants (14 females and 18 males, 3 left-handed and 29 right-handed, aged 63 ± 16 (min–max: 21–79 years), with no history of neurological, orthopedic, rheumatologic, neuromuscular or visual disorders nor impairment of the mobility of their upper limb, to participate in a research study on human movement. Prior to the movement analysis, every participant underwent a standard clinical assessment of both upper limbs to verify that passive and active ranges of motion for both shoulders, elbows, wrists and finger joints were full and pain free. This assessment was realized by two senior PM&R specialists. We then asked every subject for their dominant upper limb and confirmed that statement when they were to sign the written consent.

### 2.1. Experimental Set-Up

Experimental set-up is illustrated in Figure 1. All participants underwent a 3D upper limb motion analysis while performing two sets (A and B) of three point-to-point movements with each UL. Participants were comfortably seated in a chair; in the starting position, they had the elbow flexed at 90 degrees resting on the table, the palm of their hand facing down on the table. The point-to-point movement consisted of reaching a target located in front at 90% of upper limb length (forward-pointing movement, FPM) and at clavicle height and then bringing the hand back to the baseline position (backward-pointing movement, BPM), all of it at comfortable speed. In set A, participants were asked to mark a short pause between FPM and BPM (not exceeding three seconds), while in set B, participants proceeded to BPM directly after FPM. The paradigm was chosen to test the effect of the transition between the forward and the backward movement on movement smoothness. Each full point-to-point movement was repeated four times, the first attempt being considered as a training movement and thus not recorded. These two sets of movements were completed consecutively with the dominant and the non-dominant UL. Combining these conditions, a total of 768 movements (24 movements per subject) were recorded and analyzed.

Upper limb movements were recorded using a 15-camera 100 Hz sampling Vicon motion capture system (Oxford Metrics, Oxford, UK). Reflective markers (14 mm) were placed on classical UL anatomical landmarks: the mid-hand marker was positioned on the third metacarpal on the back of each hand. Supplementary markers were positioned on the lateral condyle of the humerus, the head of the second and fifth metacarpal and styloid processes of the radius to detect potential artefacts and ensure visual consistency. The same investigator placed all markers in each session.

### 2.2. Data Analysis

Analyses focused on the mid-hand marker (placed on the third metacarpal, on the back of the hand). Each recorded trajectory was visually inspected twice by the same investigator to manually define the beginning and end of each movement. The start of the FPM was the first ascending point of the projection of the trajectory in the vertical direction; the end of the FPM, which was also the start of the BPM, was the outermost point of the trajectory in the postero-anterior direction. The end of the BPM was the last point of the descending trajectory in the vertical direction, just before the rebound of the hand on the table.

A 6-Hz second order low-pass Butterworth filter was applied to the trajectories before analyses, except for the SPARC which carries an in-built filter. All outcomes were calculated on Python software as the mean value of the three recorded movements in each set. First, second and third derivatives of trajectory of the mid-hand marker data were calculated, and visually checked for noise, across all three plans of space to retrieve the velocity, acceleration and jerk profiles. Trajectories and velocity profiles of the different movements are illustrated in Figure 2. The index of curvature (IoC) was computed and defined as the ratio of the arc length of the trajectory to the length of the straight line linking the first and the last movement points, as a straightness metric.

Smoothness was quantified using three temporal domain metrics (NARJ, LDLJ and N0C) and a frequency domain metric (SPARC). The SPARC and the LDLJ were computed using the Python code provided by Balasubramanian et al. [13], in which we added the computation of the NARJ [14] and the N0C.

#### 2.2.1. NARJ

The NARJ is the normalized version of the average rectified jerk (ARJ) previously described by Cozens and Bhakta [14]. Considering a movement of total duration *T*, with a trajectory for the *x*th degree of freedom represented by trajectory=x(t) where 0 ≤ *t* ≤ *T*, then the ARJ for the *x*th degree of freedom for this movement is given by the formula:$$\mathrm{ARJ}=\frac{1}{T}\int_{0}^{T}\left|\frac{d^{3}x\left(t\right)}{dt^{3}}\right|dt$$

The ARJ is thus highly dependent on movement duration, with a longer movement yielding a higher ARJ than a movement of identical shape but shorter duration. The NARJ was then calculated by normalizing the ARJ to a standard movement time of one second allowing the comparison of smoothness among a group of movements with the same trajectory but different durations (*T*).
NARJ_x_ = T^3^·ARJ_x_

As the jerk reflects the rate of change in the acceleration profile, an increased NARJ magnitude reflects decreased smoothness (more frequent changes in acceleration).

We are aware that jerk derived measures account for the duration of a given movement when the appropriate scaling factor is applied, but do not account for the amplitude of this movement [9]; this could be an issue when jerk-derived measures are used in clinical practice to assess movements with distances that are not normalized, as in the current study. As the NARJ was used in various studies assessing movement of neurologically impaired patients, we wanted to further explore its behavior and applicability in unconstrained movements.

#### 2.2.2. LDLJ

The LDLJ results from the logarithm naturalis of the sum of the squared acceleration multiplied by the trial duration to the power of three and divided by the squared peak velocity.
$$\mathrm{DLJ}\;≜-\frac{{\left(t2-t1\right)}^{5}}{v^{2}peak}\int_{t_{1}}^{t_{2}}{\left|\frac{d^{2}v\left(t\right)}{dt^{2}}\right|}^{2}dt\;\mathrm{LDLJ}\;≜-\ln|\mathrm{DLJ}|$$

Calculation of the LDLJ is based on the velocity profile v within the time window *t*_1_ to *t*_2_. The LDLJ was designed to fix one major shortcoming of jerk-metrics, direct dependence to movement amplitude and duration.

#### 2.2.3. SPARC

The SPARC was computed from the arc length of the power spectrum of a Fourier transformation of the velocity signal, with (0, ω*_c_*) being the frequency band (where $\omega_{c}^{max}$ is the upper threshold of the cut-off frequency) and *V*(ω) the Fourier magnitude spectrum, as defined by Balasubramanian et al [12].
$$SPARC≜-\overset{\omega_{c}}{\underset{0}{\int }}{\left[{\left(\frac{1}{\omega_{c}}\right)}^{2}+{\left(\frac{d\hat{V}\left(\omega \right)}{d\omega }\right)}^{2}\right]}^{\frac{1}{2}}d\omega ;\;\;\;\;\;\hat{V}\left(\omega \right)=\frac{V\left(\omega \right)}{V\left(0\right)}$$
$$\omega_{c}≜min\left\{\omega_{c}^{max},\;min\{\omega ,\hat{V})<\bar{\;\forall \;r>\omega }\right\}$$

As recommended by Balasubramanian et al. [12], our choice for was 0.05, with 20 Hz for $\omega_{c}^{max}$.

#### 2.2.4. N0C

The N0C is simply defined as the number of maxima in the velocity profile. It is theoretically equal to 1 in a movement considered as smooth.

### 2.3. Statistics

Descriptive statistics were performed to provide average values with standard deviations. As data were not normally distributed (visual inspection of distribution and Shapiro-Wilk tests), non-parametrical Mann-Whitney tests for comparison were used. For each outcome, dominant and non-dominant sides, FPM and BPM and sets A and B were compared.

Validity assessment is required for each clinical measure routinely used but is rarely studied. It is an on-going process that needs successive refinements to reinforce or refute the quality of the clinical or biomechanical measures used in various conditions. Jerk derivative measures (including the ARJ and NARJ [5]) were already used in clinical studies [17,18,19], but their validity was challenged by previous studies [9,12,13], and we chose to include it in this work to bring forth new evidence to discuss its relevance.

Construct validity was evaluated using non-parametrical Spearman correlations between smoothness metrics (convergent validity) and with movement duration (divergent validity). Coefficients of variation (CoV—defined as the standard deviation divided by the mean value; used to evaluate data dispersion) were calculated between tries (CoV_intra_) and between subjects (CoV_inter_) to estimate within-and between-subject variability, respectively. All statistical analyses were performed using SPSS 20 (IBM, Armonk, NY, USA).

## 3. Results

A total of 768 movements (24 movements per subject) were analyzed. There were no missing data.

### 3.1. Usual Kinematics Parameters

All results of the usual four kinematic parameters—movement duration, index of curvature, peak velocity and mean velocity—are presented in Table 1 and Table 2. Notably, there was no systematic difference between dominant and non-dominant sides. More variation was observed across subjects than across repetitions. In set B (without pause), both movement types were shorter in duration, straighter and faster with a higher mean and peak velocity compared to set A (with pause). In both sets, BPM were shorter in duration, faster with a higher mean velocity but not straighter.

### 3.2. Smoothness Parameters

All results of the 4 smoothness parameters—N0C, NARJ, LDLJ and SPARC—are presented in Table 3 and Table 4 and Figure 3. According to the temporal domain smoothness metrics (TDSM, including N0C, NARJ and LDLJ), BPM were smoother than FPM. According to the SPARC, they were less smooth. TDSM found movements without pause to be smoother, while SPARC did not find changes in smoothness between the two sets of movements. No metric indicated differences across sides (dominant/non-dominant). Inter- and intra-subject coefficients of variation were lowest for the SPARC.

### 3.3. Correlations

Correlations of smoothness parameters are illustrated in Figure 4. The three TDSM were strongly correlated together and to movement duration. SPARC was not correlated to TDSM or to movement duration. Smoothness metrics were not correlated to age or sex.

## 4. Discussion

This study reports the comparison of the main smoothness metrics from temporal domain and the SPARC from frequency domain on different sets (with or without pause) and types of actual point-to-point movements (forward and backward) in healthy subjects. The 4 smoothness parameters, N0C, NARJ, LDLC and SPARC, were sensitive to movement type, but the SPARC behaved differently than the temporal domain smoothness metrics (TDSM), finding backward movements to be less smooth than forward movements instead of the opposite. Only the TDSM were sensitive to movement set. TDSM strongly correlated with movement duration, whereas the SPARC did not. Within-subject repeatability was highest and between-subject variability was lowest for the SPARC. No clear difference in movement velocity or smoothness was found across sides, but movements tended to be slightly straighter with the dominant arm. Mean values were reported for each metric and are now available.

### 4.1. Different Behavior between Temporal Domain Smoothness Measures and the SPARC—To Be or Not to Be Time-Connected?

The first striking finding in this study is that temporal domain smoothness measures found backward movements to be smoother than forward movements. There was only about one zero crossing on the acceleration profile in backward movements without pause while the SPARC found backward movements to be less smooth. Except for the fact that backward movements were faster than forward movements (movement time-dependence of metrics measured in the temporal domain), this opposite behavior between the two types of metrics is challenging to interpret.

Interestingly, TDSM were strongly correlated amongst them, while the SPARC was not correlated with any other metrics (low convergent validity) nor to movement duration (strong divergent validity). Additionally, TDSM were strongly correlated with movement time. Thus, the changes in smoothness perceived by TDSM across sets (with versus without pause) might reflect the sensitivity of those metrics to movement duration. From mathematical models, Balasubramanian suggested shortcomings for TDSM, precisely including their sensitivity to movement duration (and to noise in measurements) with a risk of lack of reliability and validity [12]. Thus, the correlation between movement duration and the TDSM might have a simple explanation based on Balasubramanian et al. 2015. In the subsection titled „Same noise results in less smoothness for slow movements”, they showed that the LDLJ is sensitive to movement duration when the noise from the measuring instrument is the same. Faster movements then mean a higher signal-to-noise ratio, and thus better smoothness. This might also explain why the backward movements were smoother with the TDSM. Another explanation would be that trajectories of backward movements were slightly different from those of forward movements, as shown in the Figure 2. Considering the trajectory profiles, backward movements seem to end more abruptly than forward movement, with no incurvation or plateau seen at the end of the movement. Moreover, standard deviations of velocity profiles appear to be wider on backward movements than on forward movement, suggesting a more important variability and thus loss of smoothness.

Intellectually though, it may be tempting to speculate that a movement that loses smoothness should also lose speed in the process. It may therefore seem natural that a smoothness metric be somewhat correlated with movement time, at least in pathological or complex movements. However, correlations between movement duration and smoothness measures have, to our knowledge, not been explored yet. Similar studies should thus be conducted in people with neurological impairment, such as post-stroke spastic paresis.

### 4.2. Coherence of the Present Smoothness Data with Previous Literature

For forward movements, the present SPARC values (−1.43 to −1.45 ± 0.03) were consistent with recent findings: −1.45 to −1.48 for Engdahl et al. [20] and −1.44 ± 0.04 for Saes et al. [21] in reach-to-grasp movements. Although the movement type was different in those studies (reach-to-grasp vs. reach-to-point), the reaching phase was predominant in both types of movement, allowing the comparison. As for LDLJ values, the present data are also consistent with those reported by Engdahl et al. [20]. The N0C values in our findings (3 to 6 ± 3) seem relatively high, as only one zero-crossing per movement is expected in healthy and even in some pathologic movements [22,23]. As for the present NARJ values, they were also consistent with previous literature [5] and varied more than twofold (20193 to 46378 ± 17085) depending on the movement type, possibly reflecting both noise and movement duration (when movements stopped during the plateau phase) sensitivity of those metrics.

Similar observations can be made for backward movements, even though we could not find data in the literature for comparisons. SPARC values were also remarkably consistent across sets (−1.48 ± 0.06), LDLJ shows some differences (−6.64 ± 0.52 to −5.58 ± 0.53), whereas NARJ and N0C values again ranged from single to double depending on the movement type.

### 4.3. Inter- and Intra-Subject Reliability and Sensitivity to Change of Smoothness Metrics

Can smoothness characterize one individual’s movement, or can it change between two different movements made by one person? Among the smoothness metrics tested here, SPARC showed the least intra- and between-subject variability for all tested movements (e.g., FPM CoV_inter_ 1.6 to 2.2%) followed by the LDLJ (with higher CoV_inter_ 6.8 to 8.7%). On the other hand, the NARJ and N0C were characterized by high levels of inter-subject differences, i.e., high CoV_inter_. Similarly, the SPARC was characterized by higher intra-subject repeatability (FPM CoV_intra_ 1.6 to 1.9%) than TDSM (FPM CoV_intra_ 5.0 to 6.4% for the LDLJ, 18 to 30% for the NARJ and 27 to 40% for the N0C).

In our protocol, each movement was repeated four times, the first attempt being considered as training and thus not retained for computation. We assumed this number of repetitions to be sufficient in healthy subjects for the SPARC computation. Indeed, no differences were observed in the literature when using 7 or 10 repetitions [21,24]. For TDSM, due to their greater variability/sensitivity to changes in movements and/or in movement durations, a higher number of repetitions might have to be used to obtain a reliable mean smoothness value, such as 7–8 movements [5]. This lower number of repetitions needed for the SPARC computation can be interesting in very impaired subjects for whom only fewer movement repetitions may be possible [24].

### 4.4. Smoothness and Laterality—Is There an ‘Optimal’ Smoothness?

If movements on the dominant side were slightly straighter, as previously described [25], we did not find smoothness differences between sides. Reaching movements are relatively simple, largely used, thus trained, in most daily activities. In more complex tasks such as transferring an object with chopsticks, movements have been shown to be longer and less smooth on the non-dominant side, but trainable to become as smooth as on the dominant side [26]. Finally, in healthy subjects, simple movements are expected to be optimally smooth; thus, a ceiling effect in data distribution was observed for the SPARC, the NARJ and the N0C. By applying a logarithmic transformation to normalize the dimensionless jerk, the LDLJ displays an artificially normal distribution, which might make interpretations of changes in smoothness more difficult than it is with the other metrics.

### 4.5. Study Limitations

Subjects were asked to complete each task at their preferred speed and yet differences in movement durations across movement types and sets were found, suggesting an explanation for changes in smoothness reported by TDSM. It would have been useful to add sets of movements of various imposed speeds, including ballistic movements at maximal speed, to better explore sensitivity of the four metrics to movement duration [5].

Reach-to-point movements were tested to aim for normative smoothness data that could be of interest in clinical routine. Pointing movements requiring multi-joint coordination are often used in clinical practice, especially in neurorehabilitation to assess patient progression, as these are simple to assess, reproducible [27] and frequently used in daily living activities [28]. Moreover, reach-to-point movements are more easily performed than reach-to-grasp movements, especially in numerous neurological disorders that can prevent patients from grasping (stroke, advanced parkinsonism, severe peripheral neuropathies, myopathies…) and have been recommended in more impaired patients [4]. However, smoothness of other movements of interest still needs to be quantified, especially of single-joint movements with the hope of better differentiation between recovery and compensation [21].

To date, there is no clear consensus on how to determine with precision the onset and the end of a reaching movement. As explained in the Methods section, a single assessor visually inspected each recorded trajectory twice and standardized the beginnings of FPM as the first ascending point of the projection of the trajectory in the vertical direction, and the end of FPM (which also was the beginning of the BPM) as the most forward point of the trajectory in the antero-posterior direction; the end of the BPM being the last point of the descending trajectory in the vertical direction. This method is time-consuming and might have resulted in errors, in contrast with other methods that rely on the detection of changes in maximum tangential speed during the various phases of the movement [21,28,29]. However, those definitions for the starting and ending points of trajectories allow considering more trajectory points at critical stages of movements (such as when nearing the target, where accuracy is needed), with thus perhaps more accurate estimations of smoothness across the whole movement.

## 5. Conclusions

This study of actual movements (rather than models) reveals major differences between smoothness metrics measured in the temporal domain (N0C, LDLJ, NARJ), strongly correlated amongst themselves and with movement duration and the SPARC, derived from the frequency domain. Among smoothness metrics, the SPARC showed less between-subject and within-subject variability for all tested movements and revealed changes between forward and backward movement that were opposite to time-dependent metrics. The present study also helped provide a normative dataset of smoothness measures for reaching upper limb movements in healthy subjects for each metric, which is available for future studies.

Further studies assessing these measures of smoothness on different movements of interest in upper and lower limbs are needed in healthy subjects, as well as in progressive neurological disorders such as spastic paresis syndromes or parkinsonism.

## Figures and Tables

**Figure 1 sensors-23-01158-f001:**
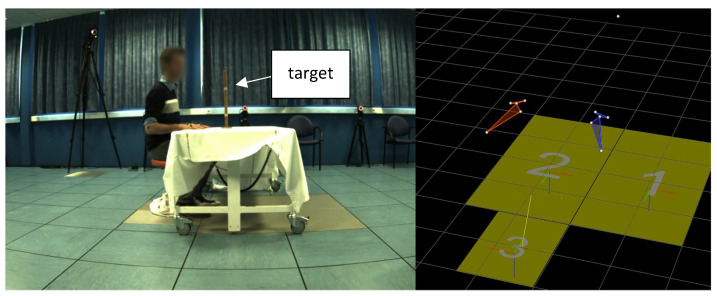
Experimental set up: starting position and target (**left**). Representation of reflective markers in Vicon software (**right**).

**Figure 2 sensors-23-01158-f002:**
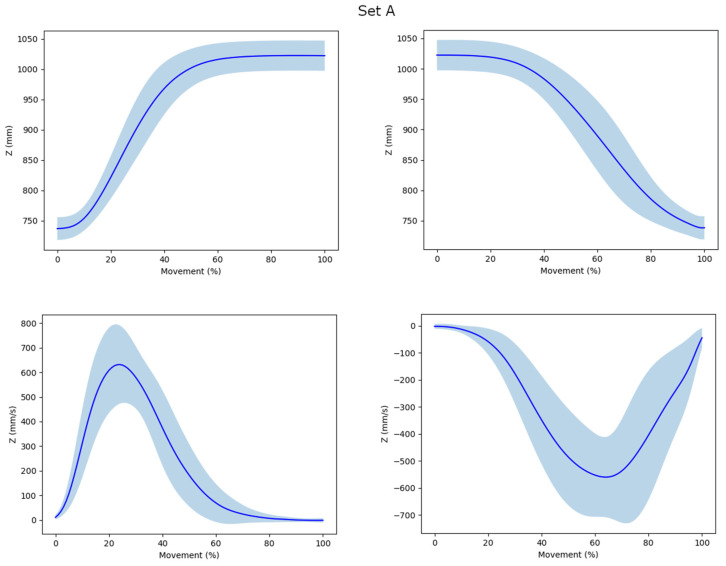
Mean value and standard deviation of trajectories (first line) and velocity profiles (second line) for set (**A**) (with pause) and set (**B**) (without pause) forward- and backward-pointing movements in the vertical plane (Z) as a function of movement completion (%).

**Figure 3 sensors-23-01158-f003:**
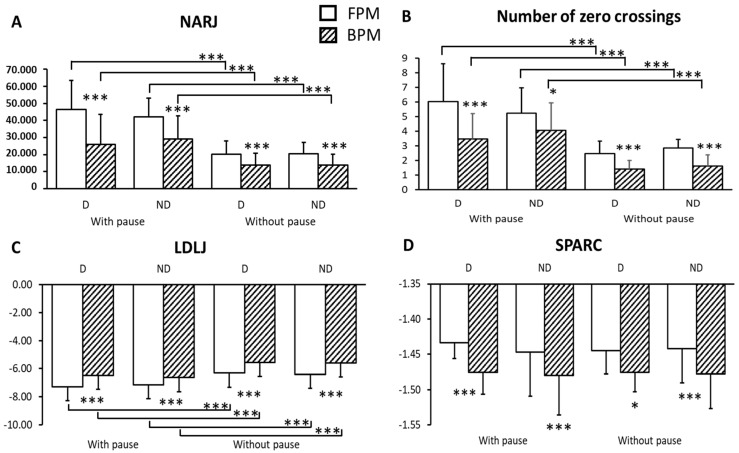
(**A**) Differences in smoothness measured with the NARJ across sides, sets (with pause vs. without pause) and movement type (forward vs. backward). (**B**) Differences in smoothness measured with the Number of Zero Crossings across sides, sets and movement type. (**C**) Differences in smoothness measured with the LDLJ across sides, sets and movement type. (**D**) Differences in smoothness measured with the SPARC across sides, sets and movement type; (Mann–Whitney tests). FPM: forward-pointing movement; BPM: backward-pointing movement; D: dominant, ND: non-dominant; * *p* < 0.05; *** *p* < 0.01.

**Figure 4 sensors-23-01158-f004:**
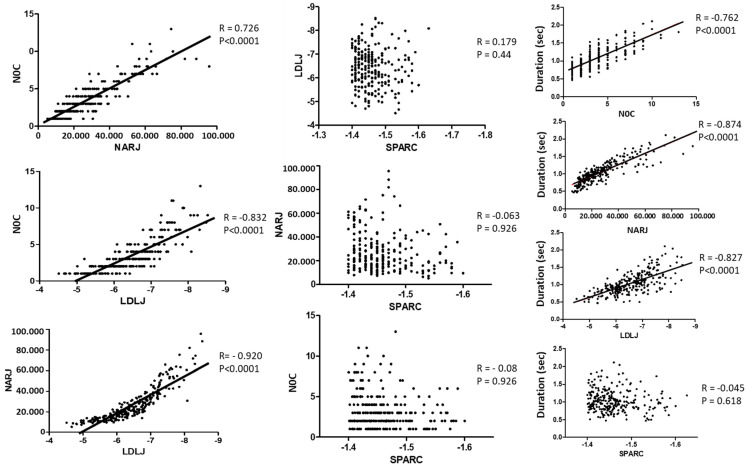
First column: Spearman correlations between temporal domain smoothness metrics. Second column: Spearman correlations between SPARC and temporal domain metrics. Third column: Spearman correlations between smoothness metrics and movement duration.

**Table 1 sensors-23-01158-t001:** Movement set comparisons for a given direction.

	**KINEMATICS—Movement Sets Comparisons**									
			**Time (s)**		**Index of Curvature**		**Velocity (Peak—cm/s)**		**Acc (Peak—cm/s^2^)**	
**A**	**FPM**		**D**	**ND**	**D**	**ND**	**D**	**ND**	**D**	**ND**
	**With Pause (A)**	**Mean (SD)**	1.40 (0.28)	1.31 (0.26) *	5.2 (2.1)	6.0 (2.9) *	84.8 (15.3)	86.0 (19.3)	462.0 (165.6)	486.6 (209.2)
		**CoV inter**	20.4	20.1	40.2	48.4	18.0	22.4	35.9	43.0
		**CoV intra**	11.4	13.7	21.5	25.9	6.6	7.1	12.8	14.4
	**W/o pause (B)**	**Mean (SD)**	0.92 (0.18)	0.92 (0.20)	4.4 (2.1)	5.1 (2.2)	92.5 (18.0)	93.7 (22.0)	542.1 (201.6)	566.4 (245.6)
		**CoV inter**	20.1	21.7	47.3	41.9	19.5	23.5	37.2	43.4
		**CoV intra**	8.5	9.1	30.7	30.4	5.9	6.4	15.2	15.5
		** *p* **	<0.000001	0.000001	0.007	0.045	0.00008	0.00004	0.0006	0.0001
			**Time (s)**		**Index of Curvature**		**Velocity (Peak—cm/s)**		**Acc (PEAK—cm/s^2^)**	
**B**	**BPM**		**D**	**ND**	**D**	**ND**	**D**	**ND**	**D**	**ND**
	**With Pause (A)**	**Mean (SD)**	1.06 (0.23)	1.10 (0.25)	5.5 (2.4)	6.5 (3.7)	84.0 (20.1)	81.7 (18.6)	361.4 (151.2)	372.8 (157.2)
		**CoV inter**	21.6	22.3	44.0	56.6	23.9	22.8	41.8	42.2
		**CoV intra**	14.1	14.6	28.4	29.6	9.2	7.8	19.6	19.9
	**W/o pause (B)**	**Mean (SD)**	0.80 (0.18)	0.80 (0.21)	4.5 (2.7)	5.3 (3.0) *	88.9 (20.8)	88.5 (22.0)	386.6 (139.4)	405.7 (172.8)
		**CoV inter**	22.6	25.9	59.1	57.2	23.4	24.9	36.1	42.6
		**CoV intra**	9.3	8.5	27.5	25.3	8.5	8.6	15.8	17.5
		** *p* **	<0.000001	<0.000001	0.04	0.02	0.07	0.005	0.24	0.08

Healthy values of kinematic parameters [mean(SD)] across sets. (**A**) Comparison between sets (with or without pause) of the kinematics of FPM. (**B**) Comparison between sets (with or without pause) of the kinematics of BPM Mann–Whitney tests were used to compute *p*-values. FPM: forward-pointing movement; BPM: backward-pointing movement; CoV intra: coefficient of variation calculated between tries; CoV inter: coefficient of variation calculated between subjects; Acc: acceleration; D: dominant, ND: non-dominant; * *p* < 0.05, D vs. ND.

**Table 2 sensors-23-01158-t002:** Movement direction comparisons across sets.

	**KINEMATICS—Movement Direction Comparisons**									
			**Time (s)**		**Index of Curvature**		**Velocity (Peak—cm/s)**		**Acc (Peak—cm/s^2^)**	
**C**	**With Pause (A)**		**D**	**ND**	**D**	**ND**	**D**	**ND**	**D**	**ND**
	**FPM**	**Mean (SD)**	1.40 (0.28)	1.31 (0.26) *	5.2 (2.1)	6.0 (2.9) *	84.8 (15.3)	86.0 (19.3)	462.0 (165.6)	486.6 (209.2)
		**CoV inter**	20.4	20.1	40.2	48.4	18.0	22.4	35.9	43.0
		**CoV intra**	11.4	13.7	21.5	25.9	6.6	7.1	12.8	14.4
	**BPM**	**Mean (SD)**	1.06 (0.23)	1.10 (0.25)	5.5 (2.4)	6.5 (3.7)	84.0 (20.1)	81.7 (18.6)	361.4 (151.2)	372.8 (157.2)
		**CoV inter**	21.6	22.3	44.0	56.6	23.9	22.8	41.8	42.2
		**CoV intra**	14.1	14.6	28.4	29.6	9.2	7.8	19.6	19.9
		** *p* **	<0.000001	0.001	0.54	0.4	0.68	0.14	0.0006	0.0003
			**Time (s)**		**Index of Curvature**		**Velocity (Peak—cm/s)**		**Acc (Peak—cm/s^2^)**	
**D**	**W/o Pause (B)**		**D**	**ND**	**D**	**ND**	**D**	**ND**	**D**	**ND**
	**FPM**	**Mean (SD)**	0.92 (0.18)	0.92 (0.20)	4.4 (2.1)	5.1 (2.2)	92.5 (18.0)	93.7 (22.0)	542.1 (201.6)	566.4 (245.6)
		**CoV inter**	20.1	21.7	47.3	41.9	19.5	23.5	37.2	43.4
		**CoV intra**	8.5	9.1	30.7	30.4	5.9	6.4	15.2	15.5
	**BPM**	**Mean (SD)**	0.80 (0.18)	0.80 (0.21)	4.5 (2.7)	5.3 (3.0) *	88.9 (20.8)	88.5 (22.0)	386.6 (139.4)	405.7 (172.8)
		**CoV inter**	22.6	25.9	59.1	57.2	23.4	24.9	36.1	42.6
		**CoV intra**	9.3	8.5	27.5	25.3	8.5	8.6	15.8	17.5
		** *p* **	0.00003	0.001	0.9	0.78	0.12	0.08	0.00001	0.000001

Healthy values of kinematic parameters [mean(SD)] across movements directions. (**C**) Comparison of the kinematics between FPM and BPM in set A setting (with a pause). (**D**) Comparison of the kinematics between FPM and BPM in set B setting (without a pause). Mann–Whitney tests were used to compute *p*-values. FPM: forward-pointing movement; BPM: backward-pointing movement; CoV intra: coefficient of variation calculated between tries; CoV inter: coefficient of variation calculated between subjects; Acc: acceleration; D: dominant, ND: non-dominant; *: *p* < 0.05, D vs. ND.

**Table 3 sensors-23-01158-t003:** Movement set comparisons for a given direction.

	**SMOOTHNESS—Movement Sets Comparisons**									
			**N0C**		**NARJ × 10^3^ (mm/s^3^)**		**LDLJ**		**SPARC**	
**A**	**FPM**		**D**	**ND**	**D**	**ND**	**D**	**ND**	**D**	**ND**
	**With Pause (A)**	**Mean (SD)**	6 (3)	5 (2)	46.4 (17.1)	42.1 (17.5)	−7.29 (0.50)	−7.14 (0.62)	−1.43 (0.02)	−1.45 (0.03) *
		**CoV inter**	42.9	46.8	36.8	41.5	6.8	8.7	1.6	2.1
		**CoV intra**	36.4	39.7	26.0	30.1	5.1	6.4	1.6	1.9
	**W/o pause (B)**	**Mean (SD)**	3 (1)	3(1) *	20.2 (7.9)	20.5 (6.9)	−6.32 (0.53)	−6.41 (0.44)	−1.45 (0.03)	−1.44 (0.03)
		**CoV inter**	34.6	32.2	39.3	33.8	8.4	6.9	2.2	1.9
		**CoV intra**	26.8	29.3	18.1	21.1	5.0	5.9	1.9	1.8
		** *p* **	0.000002	0.000006	0.000001	0.000001	<0.000001	<0.000001	0.075	0.259
			**N0C**		**NARJ × 10^3^ (mm/s^3^)**		**LDLJ**		**SPARC**	
**B**	**BPM**		**D**	**ND**	**D**	**ND**	**D**	**ND**	**D**	**ND**
	**With Pause (A)**	**Mean (SD)**	3 (2)	4 (2)	26.1 (10.9)	29.2 (13.6)	−6.48 (0.64)	−6.64 (0.52)	−1.48 (0.06)	−1.48 (0.06)
		**CoV inter**	49.5	46.4	41.7	46.7	9.8	7.8	4.2	3.8
		**CoV intra**	50.2	52.8	33.2	32.1	8.2	6.6	3.2	3.2
	**W/o pause (B)**	**Mean (SD)**	1 (1)	2 (1) *	13.9 (6.6)	13.9 (6.1)	−5.56 (0.52)	−5.58 (0.53)	−1.48 (0.05)	−1.48 (0.05)
		**CoV inter**	41	46.8	47.0	43.8	9.4	9.4	3.3	3.3
		**CoV intra**	24.9	36.3	20.7	21.3	5.8	5.6	3.1	3.5
		** *p* **	0.000003	0.000002	0.000002	0.000001	<0.000001	<0.000001	0.422	0.779

Healthy values of smoothness metrics [mean(SD)] across sets. (**A**) Comparison between sets (with or without pause) of smoothness metrics of FPM. (**B**) Comparison between sets (with or without pause) of smoothness metrics of BPM Mann-Whitney tests were used to compute *p*-values. FPM: forward-pointing movement; BPM: backward-pointing movement; CoV intra: coefficient of variation calculated between tries; CoV inter: coefficient of variation calculated between subjects; D: dominant, ND: non-dominant; * *p* < 0.05, D vs. ND.

**Table 4 sensors-23-01158-t004:** Movement direction comparisons across sets.

	**SMOOTHNESS—Movement Direction Comparisons**									
			**N0C**		**NARJ × 10^3^ (mm/s^3^)**		**LDLJ**		**SPARC**	
**C**	**With Pause (A)**		**D**	**ND**	**D**	**ND**	**D**	**ND**	**D**	**ND**
	**FPM**	**Mean (SD)**	6 (3)	5 (2)	46.4 (17.1)	42.1 (17.5)	−7.29 (0.50)	−7.14 (0.62)	−1.43 (0.02)	−1.45 (0.03) *
		**CoV inter**	42.9	46.8	36.8	41.5	6.8	8.7	1.6	2.1
		**CoV intra**	36.4	39.7	26.0	30.1	5.1	6.4	1.6	1.9
	**BPM**	**Mean (SD)**	3 (2)	4 (2)	26.1 (10.9)	29.2 (13.6)	−6.48 (0.64)	−6.64 (0.52)	−1.48 (0.06)	−1.48 (0.06)
		**CoV inter**	49.5	46.4	41.7	46.7	9.8	7.8	4.2	3.8
		**CoV intra**	50.2	52.8	33.2	32.1	8.2	6.6	3.2	3.2
		** *p* **	0.0001	0.033	0.00001	0.004	0.000003	0.001	0.002	0.009
			**N0C**		**NARJ × 10^3^ (mm/s^3^)**		**LDLJ**		**SPARC**	
**D**	**W/o Pause (A)**		**D**	**ND**	**D**	**ND**	**D**	**ND**	**D**	**ND**
	**FPM**	**Mean (SD)**	3 (1)	3(1) *	20.2 (7.9)	20.5 (6.9)	−6.32 (0.53)	−6.41 (0.44)	−1.45 (0.03)	−1.44 (0.03)
		**CoV inter**	34.6	32.2	39.3	33.8	8.4	6.9	2.2	1.9
		**CoV intra**	26.8	29.3	18.1	21.1	5.0	5.9	1.9	1.8
	**BPM**	**Mean (SD)**	1 (1)	2 (1) *	13.9 (6.6)	13.9 (6.1)	−5.56 (0.52)	−5.58 (0.53)	−1.48 (0.05)	−1.48 (0.05)
		**CoV inter**	41	46.8	47.0	43.8	9.4	9.4	3.3	3.3
		**CoV intra**	24.9	36.3	20.7	21.3	5.8	5.6	3.1	3.5
		** *p* **	0.000004	0.00001	0.000004	<0.00001	<0.000001	<0.000001	0.011	0.001

Healthy values of smoothness metrics [mean(SD)] across movements directions. (**C**) Comparison of the smoothness metrics between FPM and BPM in set A setting (with a pause). (**D**) Comparison of the smoothness metrics between FPM and BPM in set B setting (without a pause). Mann–Whitney tests were used to compute *p*-values. FPM: forward-pointing movement; BPM: backward-pointing movement; CoV intra: coefficient of variation calculated between tries; CoV inter: coefficient of variation calculated between subjects; D: dominant, ND: non-dominant; * *p* < 0.05, D vs. ND.

## Data Availability

The corresponding author of this article will share all data that underlie the results reported in this article with qualified researchers who provide a valid research question.

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
