# Peer review of "Comparison of Various Smoothness Metrics for Upper Limb Movements in Middle-Aged Healthy Subjects"

_sensors, 2023, doi:10.3390/s23031158_

Round 1
Reviewer 1 Report
In this manuscript, comparison of various smoothness metrics for upper limb movements in healthy subjects was reported. The comparison is meaningful, and is interested for readers. However, there exist some problems in this manuscript.
1. The format of the equation is not standard.
2. The layout of Table 2 should be adjusted.
Therefore, I recommend it to be published after minor revision.
Reviewer 2 Report
1. I noticed there is a preprint of your manuscript on Research Square. https://doi.org/10.21203/rs.3.rs-1431894/v1
2. Please prepare the manuscript carefully. Address 4 is missing.
3. In part discussion, the author claimed "This study reports for the first time, at our knowledge, the comparison of the main 249 smoothness measures from both temporal and frequency domains on different sets (with 250 or without pause) and types of actual point-to-point movements (forward and backward) 251 in healthy subjects." But is it really the case?
4. I didn't find any serious shortcomings in the manuscript.
Reviewer 3 Report
The idea of the paper is very intersecting, and I was very enthusiastic. Unfortunately, the manuscript has many shortcomings and needs to be improved. Among other things, the purpose of the study is to build normative values. I have not seen them. Please make it clear where they are. In addition, I have comments on the results section, which is not clear and understandable because the methods do not describe what is compared with what. Please, find detailed comments below.

Round 2
Reviewer 3 Report
The authors responded to all my comments. I do not have any further ones. I like the paper very much.